# Wrist Circumference Cutoff Points for Determining Excess Weight Levels and Predicting Cardiometabolic Risk in Adults

**DOI:** 10.3390/ijerph21050549

**Published:** 2024-04-26

**Authors:** Larissa Monteiro Costa Pereira, Márcia Ferreira Cândido de Souza, Felipe J. Aidar, Márcio Getirana-Mota, Alex Menezes dos Santos-Junior, Mario Francisco Dantas de Santana Filho, Marcos Antonio Almeida-Santos, Raysa Manuelle Santos Rocha, Rebeca Rocha de Almeida, Leonardo Baumworcel, Luiz Henrique Sala de Melo Costa, Renata Rebello Mendes, Antônio Carlos Sobral Sousa

**Affiliations:** 1Graduate Program in Health Sciences, Federal University of Sergipe (UFS), Aracaju 49100-676, Brazil; larissa_monteiroo@hotmail.com (L.M.C.P.); ysamanu@hotmail.com (R.M.S.R.); rebeca_nut@hotmail.com (R.R.d.A.); acssousa@terra.com.br (A.C.S.S.); 2Graduate Program in Nutritional Sciences, Federal University of Sergipe (UFS), São Cristóvão 49100-000, Brazil; nutrimarciacandido@gmail.com (M.F.C.d.S.); alex_jrmenezes@live.com (A.M.d.S.-J.); mariofranciscods1@gmail.com (M.F.D.d.S.F.); 3Graduate Program in Physiological Sciences, Federal University of Sergipe (UFS), São Cristóvão 49100-000, Brazil; marcio_getirana@hotmail.com; 4Graduate Program in Physical Education, Federal University of Sergipe (UFS), São Cristóvão 49100-000, Brazil; 5Graduate Program in Health and Environment, Tiradentes University-UNIT, Aracaju 49032-490, Brazil; marcosalmeida2010@yahoo.com.br; 6Division of Cardiology, University Hospital of Federal University of Sergipe (UFS), Aracaju 49100-000, Brazil; leonardo.baumworcel@caxiasdor.com.br; 7Clinic and Hospital São Lucas/Rede D’Or São Luiz, Aracaju 49060-676, Brazil; costa.luizhenrique@hotmail.com; 8Department of Nutrition, Federal University of Sergipe (UFS), São Cristóvão 49100-000, Brazil; remendes@academico.ufs.br; 9Department of Medicine, Federal University of Sergipe (UFS), Aracaju 49100-000, Brazil

**Keywords:** adults with excess weight and predicted cardiometabolic risk

## Abstract

(1) Background: An elevated wrist circumference may indicate excess weight and cardiometabolic risk. The present study aims to identify wrist circumference cutoff points (WrC) to determine excess weight levels and predict cardiometabolic risk in adults. (2) Methods: A cross-sectional study was conducted with adults aged 20 to 59 years old, attending the outpatient clinic at University Hospital/Federal University of Sergipe HU/UFS-EBSERH. Demographic, anthropometric, biochemical, and blood pressure (BP) data were collected. Cardiometabolic risk was assessed, according to the global risk score (ERG) and Framingham score criteria. The descriptive analysis included calculating medians and frequencies of anthropometric, demographic, biochemical, and blood pressure variables. The gender and age of adult groups were compared using the Mann–Whitney test. Spearman’s correlation coefficient and multiple regression analysis were used to assess the association between wrist circumference (WrC) and the variables mentioned above. The predictive validity of WrC in identifying excess weight levels and cardiometabolic risk was analyzed using the ROC curve. The sample consisted of 1487 adults aged 20 to 59 years, 55.7% of whom were female; (3) Results: WrC correlated positively with other adiposity indicators such as waist circumference and Body Mass Index. WrC was the anthropometric indicator most significantly associated with cardiometabolic risk factors. WrC cutoff points identified by the study for determining excess weight were categorized by gender and age group. For males aged 20 to 40 years and >40 years, respectively, the cutoff points for overweight were 17.1 cm and 17.3 cm, and for obesity, 17.9 cm and 17.5 cm. For females aged 20 to 40 years and >40 years, respectively, the cutoff points for overweight were 15.6 cm and 15.4 cm, and for obesity, 16.1 cm and 16 cm (4). Conclusions: Wrist circumference showed a significant correlation with other adiposity indicators and can be used to identify adults with excess weight and predict cardiometabolic risk.

## 1. Introduction

Obesity is a major public health issue, particularly in developing countries, such as Brazil, with multifactorial causes [1,2,3], and is associated with the risk of cardiovascular diseases (CVD). Early diagnosis and control of cardiovascular risk factors are essential to reduce the prevalence of excess weight [4]. Anthropometric data are effective indirect methods to detect risk factors for obesity and CVD [5,6]. They are used in clinical and epidemiological studies [7]. Also, they are easily measured and cost-effective [8]. However, there are limitations associated with each of these indices and dimensions, so they may not always be considered adequate measures [9,10].

Wrist circumference (WrC) is a recent parameter for assessing body fat [11], and serves as a simple anthropometric tool for measuring skeletal size [12]. WrC is also used to evaluate body fat, as it is a practical, standardized measure that is not influenced by postprandial abdominal distension or respiratory movements, providing consistent results to indicate subcutaneous fat accumulation in the upper body and excess body fat [13,14]. In a meta-analysis, Namazi et al. [10], concluded that the lack of available cutoff points for WrC limits its use as an anthropometric index in clinical settings. There are insufficient data in the literature to evaluate the eligibility of WrC as a risk marker for CVD [15]. The present study aims to identify the wrist circumference cutoff points that determine excess weight levels and predict cardiometabolic risk in adults.

## 2. Materials and Methods

### 2.1. Study Design

This is a cross-sectional, observational, retrospective, and analytical investigation that adhered to the Strengthening the Reporting of Observational Studies in Epidemiology (STROBE) protocol [16], for observational studies (Figure 1). The research included volunteers aged 20 to 59 years with non-communicable chronic diseases who were seen at the outpatient clinic and did not have confirmed cardiovascular diseases. Ethical Considerations This investigation was approved by the Research Ethics Committee of the Federal University of Sergipe, with approval number 2,928,543, granted on 2 August 2018. Only patients who met the inclusion criteria and understood and signed the Informed Consent Form (ICF) were included.

### 2.2. Study Location

Data collection took place from January 2018 to May 2023, through medical records and forms used in nutritional care for patients cared for and monitored by multidisciplinary teams. The study was carried out in the outpatient clinics of the Hospital Universitário de Sergipe–HU-UFS-EBSERH, located in Aracaju, Sergipe, Brazil, a reference in public health care.

### 2.3. Clinical Data

The collected clinical data included comorbidities, systemic blood pressure, the use and category of medications for managing metabolic conditions (DM2, HAS, DLP), symptoms, and clinical signs.

### 2.4. Biochemical

Biochemical data included serum and/or plasma measurements of triglycerides, total cholesterol, HDL, LDL, fasting glucose, and glycosylated hemoglobin (HbA1c).

### 2.5. Cardiovascular Risk

For the analysis of cardiovascular risk, the Global Risk Score (ERG) and Framingham Risk Score were used.

The ERG, adopted by the Brazilian Society of Cardiology (SBC), estimates the 10-year risk of myocardial infarction, stroke, or heart failure, whether fatal, non-fatal, or peripheral vascular insufficiency. Patients are stratified into very high, high, intermediate, and low-risk categories, both for those taking statins and those not receiving hypolipidemic treatment [17].

The Framingham risk score includes the following risk factors in its analysis: age, total cholesterol, HDL-c, systemic blood pressure, diabetes, and smoking, with specific scoring for each item. The total points assign each individual to one of the following categories: low risk (≤10%), moderate risk (>10%), and high risk (≥20%) [18].

Hypertension was defined as systolic blood pressure (SBP) ≥ 140 mmHg, and/or diastolic blood pressure (DBP) ≥ 90 mmHg. Diabetes mellitus was defined by the American Diabetes Association (ADA) criteria as HbA1c ≥ 6.5% or FPG ≥ 7.0 mmol/L or self-reported history of diabetes (ADA, 2019). Dyslipidemia was defined with at least one of the following characteristics: elevated TC (TC ≥ 6.22 mmol/L); elevated LDL-C (LDL-C ≥ 4.14 mmol/L); Low HDL-C (HDL-C < 1.04 mmol/L) or hypertriglyceridemia (TG ≥ 2.26 mmol/L).

### 2.6. Anthropometry

Anthropometric data were collected by trained professionals following a standard protocol to ensure data quality and consistency. Measured variables included weight (kg), height (cm), waist, neck, wrist, and hip circumferences. After collection, the following variables were calculated: Body Mass Index (BMI), calculated as weight (with an accuracy of 0.1 kg) in kilograms divided by height (with an accuracy of 0.1 cm) in square meters (kg/m^2^); Waist-to-Hip Ratio (WHR), calculated as waist circumference (cm) divided by hip circumference (cm); Waist-to-Height Ratio (WHR), calculated as waist circumference (cm) divided by height (cm). WrC was measured with the individual seated using a tensioned measuring tape positioned over the Lister tubercle of the distal radius and the distal ulna. The Lister tubercle, a dorsal tubercle of the radius, can be easily palpated. The dorsal aspect of the radius around the level of the ulna head is approximately 1 cm proximal to the radiocarpal joint space.

### 2.7. Sociodemographic and Lifestyle

Information was collected regarding gender, marital status, age, education, and employment status. Lifestyle was also assessed, including the analysis of alcohol consumption, smoking, and use of the International Physical Activity Questionnaire Short Form (IPAQ-SF), validated for the Brazilian population. The IPAQ-SF classified physical activity based on the number of minutes of physical activity performed per week (inactive < 150 min of physical activity per week, active ≥ 150 min of physical activity per week) [19].

### 2.8. Statistical Analysis

A variety of statistical methods were employed in this study, including chi-square tests, Mann–Whitney tests, Pearson’s Chi-Square test, and descriptive measures such as median, interquartile range, absolute frequency, and percentages. Descriptive measures were used to describe variable characteristics and provide summarized information about the collected data. The chi-square test was used to investigate the association between different categorical variables. The Shapiro–Wilk test is a statistical test used to check for normal data distribution. In this study, normality in the data was not observed. Therefore, the Mann–Whitney test was employed to compare the medians of two independent samples.

Linear regression was used to model the relationship between a dependent variable and one or more independent variables. In linear regression analysis, regression coefficients were estimated to represent the relationship between independent variables and the dependent variable, while holding other variables constant. In addition to regression coefficients, standardized coefficients were also calculated, expressing the relationship between independent variables and the dependent variable in terms of standard deviations, enabling a comparison of the relative impact of independent variables, regardless of the units of measurement used.

Standard errors are estimates of the variability of regression coefficients. The T-statistic is used to test whether a regression coefficient is statistically different from zero when it is greater than 2 or the associated *p*-value is less than the significance level. The coefficient of determination, represented by R^2^, is a measure indicating the proportion of the dependent variable’s variability explained by the independent variables. Adjusted R^2^ is a corrected version of R^2^ that takes into account the number of independent variables in the model. It penalizes the inclusion of irrelevant variables and helps prevent model overfitting.

Furthermore, to assess the assumption of multicollinearity, the Variance Inflation Factor (VIF) is commonly calculated, where values above five indicate high correlation and may suggest the need for a model review. In diagnostic accuracy, the Receiver Operating Characteristic (ROC) curve is a graphical representation that evaluates the performance of a binary classification model. It is constructed by plotting the true positive rate (Sensitivity) on the *y*-axis against the false positive rate (1—Specificity) on the *x*-axis at different model cutoff points. The Area Under the Curve (AUC) is a numerical measure summarizing the overall performance of the model. The AUC value ranges from 0 to 1, with a value closer to 1 indicating better model performance in discriminating between classes. An AUC of 0.5 suggests a model that performs at a chance level, while a value above 0.5 indicates superior performance. Therefore, Youden’s method is used to determine the optimal cutoff point in a binary classification model.

Youden’s method seeks to find the cutoff point that maximizes the sum of the model’s Sensitivity and Specificity. This cutoff point is considered the most balanced in terms of correctly classifying both true positives and true negatives. Youden’s method is calculated by identifying the point on the ROC curve that has the greatest distance from the diagonal reference line (where Sensitivity = Specificity). This point corresponds to the optimal cutoff point, which maximizes both Sensitivity and Specificity simultaneously.

To evaluate the quality of the chosen cutoff points, several diagnostic accuracy metrics were employed. The metrics used include Sensitivity, which measures the ability to detect positive cases correctly; Specificity, which measures the ability to exclude negative cases correctly; Positive Predictive Value (PPV), which measures the probability that a positive result is true; Negative Predictive Value (NPV), which measures the probability that a negative result is true; Overall Accuracy, which measures the proportion of correctly identified cases across all classes; and the Diagnostic Odds Ratio, which is the ratio of the odds of positivity in individuals with the disease compared to the odds of individuals without the disease [20].

In this study, all statistical analyses were performed using the R programming environment (version 4.2.3) (The R Foundation, Indianapolis, IN, USA), and a significance level of 5% was applied to all hypothesis tests.

## 3. Results

The study sample consisted of 1532 adult individuals, with 45 participants excluded for not completing the necessary data for analysis. The majority of the sample was composed of women (55.7%). Concerning the Framingham risk classification and ERG, high risk was more frequent among individuals over 40 years old (62.4% vs. 32%, *p* < 0.001). Regarding the classification of inactivity by IPAQ (75.1% vs. 65.7%, *p* < 0.001) and the higher prevalence of obesity according to BMI (61.8% vs. 59.3%, *p* < 0.001), they were more frequent among individuals aged 20 to 40 years. The characterization of the study sample is described in Table 1.

According to the results of anthropometric data, significant differences can be observed among age groups. Weight and height in the age group of 20 to 40 years showed higher measurements (*p* = 0.002; *p* < 0.001), and the lowest values were found for waist circumference (*p* = 0.003), as well as waist-to-height ratio (*p* < 0.001) and waist-to-hip ratio (*p* < 0.001). Although there was a significant difference in wrist circumference (*p* = 0.009) among age groups, this difference was not in the median since it was the same in both groups. It pertained to the distribution of variable values (Table 2).

Wrist circumference showed a significant correlation (*p* < 0.05) with all anthropometric indicators, both in the total sample and in the results of these measurements by age group (Table 3).

When performing multiple linear regression to assess the association of wrist circumference with other anthropometric data and systemic blood pressure, the results showed that anthropometric measurements and blood pressure had a significant impact. For each additional centimeter in wrist circumference, there was an increase of 7.12 kg (Bstd = 0.61) in weight, 0.72 cm (Bstd = 0.17) in height, 2.30 kg/m^2^ (Bstd = 0.57) in BMI, 0.84 cm (Bstd = 0.49) in neck circumference, 4.86 cm (Bstd = 0.53) in waist circumference, 0.03 (Bstd = 0.47) in waist-to-height ratio, 0.01 (Bstd = 0.19) in waist-to-hip ratio, and blood pressure measurements of 1.35 mmHg (Bstd = 0.18) for systolic blood pressure and 0.91 mmHg (Bstd = 0.20) for diastolic blood pressure, respectively. All models involving anthropometric variables had a moderate predictive capacity with R^2^ and adjusted R^2^ between 0.22 and 0.43, with no violations of the multicollinearity assumption (VIF < 5). Table 4 presents the results of the multiple linear regression.

Table 5 shows the correlation values between wrist circumference and various biochemical variables related to RCM (Risk Cardiovascular Metabolic) in the study’s sample of individuals. Wrist circumference exhibited a positive correlation with total cholesterol, triglycerides, and fasting blood glucose in both the total sample and the 20-to-40 age group, indicating a correlation between increased values of these biochemical indicators and increased wrist circumference.

The graphical results of the ROC curve to determine the sensitivity and specificity of wrist circumference in classifying excess weight among female adults in the sample show AUC values above 0.7, indicating reasonable predictive quality for most cases, except for the Framingham risk in both age groups and ERG in the >40 age group (Figure 2).

Cut-off points for Wrist Circumference (WrC) to predict excess weight, Framingham risk, and ERG for females according to age groups in the study are presented in Table 6.

The graphical results of the ROC curve for determining the sensitivity and specificity of wrist circumference in classifying excess weight among male adults in the sample show AUC values above 0.7, indicating reasonable predictive quality for most cases, except for the Framingham risk in both age groups and ERG in the >40 age group (Figure 3).

Cut-off points for Wrist Circumference (WrC) for predicting excess weight, Framingham risk, and ERG for males according to age groups in the study are presented in Figure 4.

## 4. Discussion

Adult participants in the study who were older than 40 years presented lower values for weight and height, as well as higher values for WrC, Wrist-to-Height Ratio (RCE), Waist-to-Hip Ratio (RCQ), and Blood Pressure (PA). Wrist Circumference (WrC) showed significant correlations with all anthropometric indicators and blood pressure, similar to the findings in studies by Obirikorang et al. [11], Mousapour et al. [20], Payab et al. [21], Zadeh et al. [22].

Based on these studies, WrC can be used effectively as a reproducible measure for clinical practice and epidemiological studies. It serves as an independent marker of visceral adiposity associated with adipose tissue dysfunction and can identify levels of excess weight and cardiometabolic risk to prevent cardiovascular diseases in adulthood. It becomes an efficient and cost-effective strategy to detect this nutritional disorder in public health [13,22].

In our population, the prevalence of obesity was higher in the 20-to-40 age group, as well as inactivity, when comparing groups. When evaluating the proportion of overweight (overweight and obesity), the highest prevalence was among adults over 40 years old. The high percentage of overweight adults represents significant data, emphasizing the need for early nutritional intervention to minimize health-related complications associated with obesity and comorbidities. According to data published by Vigitel [23] Aracaju was ranked as the capital city with the highest percentage of obese individuals (25%) in Brazil.

The predominance of females in the sample is characterized by the increased focus on the health of this population. Within the continuum of care, physical inactivity has been associated with obesity and a higher cardiometabolic risk. Sedentary behavior is prevalent among overweight individuals, and promoting physical activity can reduce comorbidities, serving as an effective long-term treatment. A cross-sectional study conducted in Brazil demonstrated the prevalence of physical inactivity among young Brazilian adults, similar to what was observed in our study [24].

A systematic review by Burgess et al. [25] found that the primary determinants of non-adherence to lifestyle intervention in overweight individuals or those attempting lifestyle changes were related to lack of motivation, lack of time, environmental and societal pressures, health and physical limitations, negative thoughts, socioeconomic constraints, knowledge gaps, and lack of enjoyment during exercise. Affinat et al. [26] reported that the combination of factors related to excess weight and reduced physical activity could contribute to a progressive decrease in insulin production and an increase in insulin resistance, raising the risk of comorbidities.

There was a significant correlation between WrC and total cholesterol, indicating a trend of increasing total cholesterol with increasing wrist circumference. Recent studies have positively correlated WrC with weight, BMI, Neck Circumference (WrCE), Waist Circumference (WC), RCE, and RCQ, data that align with our study. WrC was positively associated with cardiometabolic risk factors such as waist circumference, BMI, and total cholesterol but not significantly with HDL-C, as evaluated in some studies. In a cohort study, WrC and total cholesterol played significant roles in predicting cardiovascular risk from adolescence to adulthood [27].

A cohort study in Iran with young adults showed that men had a greater wrist circumference range, and worse cardiometabolic risk profiles were associated with larger wrist circumferences. WrC was associated with older age, higher BMI, larger waist circumference, higher blood pressure, lower levels of HDL-C, and higher triglyceride levels [23].

The present study identified cardiovascular risk using the SBD risk calculator and Framingham [18], with a higher prevalence of high-risk scores among older adults (>40 years). The advantage of using these scores over other risk classification methods is that they allow the identification of established or advanced risks. In this life stage, reducing cardiovascular outcomes reduces mortality, improves the quality of life, and increases patient life expectancy. On the other hand, the Framingham Heart Study’s analysis concludes that fat deposition in adipose tissue is more strongly associated with risk factors for women. The mechanisms by which these adverse effects occur due to fat deposition in women are not well defined yet. However, some authors suggest that women produce higher quantities of fatty acids than men [28].

Regarding WrC cutoff points, the study used the WHO [29] diagnostic classification for BMI as a reference. Obirikorang et al. [11] also determined WrC cutoff points for identifying overweight and obese adults based on BMI classification. Comparing the results of sensitivity and specificity, the cutoff points found in our study showed higher sensitivity and specificity compared to those found by Obirikorang et al. [11] for both males (50% versus 43%) and females (84% and 90%), respectively. The WrC values were similar to our findings, but they did not distinguish between age groups (young adults: 20 to 40 years and older adults: >40 years), and cardiovascular risk prediction was only significant in women.

Additionally, Mohebi et al. [30] conducted a cohort study with a stratified sample for obesity and identified RCM only in females, whereas Derakhshan et al. [31] conducted their study in male individuals. Studies that do not compare between genders may limit the analysis of results, as differences between genders in the association between wrist circumference and the occurrence of cardiometabolic risk factors may occur due to the effects of sex steroids and their interaction with bone metabolism and glucose homeostasis [32].

Capizzi et al. [33] assessed obese adolescents and suggested that wrist circumference can be considered in the classification of obesity for predicting cardiovascular risk. Amini et al. [34] conducted a study with 1709 diabetic patients and showed a significant positive association between wrist circumference and cardiometabolic risk factors. Therefore, wrist circumference measurement can serve as a clinically detectable marker for identifying individuals at risk for cardiometabolic disorders.

The results obtained suggest that measuring WrC is an important tool for identifying overweight and obese adults. The observed results are consistent, as greater precision in WrC classification cutoffs makes the results more representative, given the inclusion of individuals from different age groups and genders, compared to other studies. The study’s limitations include not conducting correlations with more accurate diagnostic tests and the limited number of studies available for better comparisons and discussions. Studies on wrist circumference measurement are generally conducted with specific subgroups of populations. This is the first time that a risk score has been used to determine WrC cutoff points, and further research is needed to evaluate this method in other samples. Therefore, it is recommended to conduct new studies with larger and more representative national and regional samples to discuss the results of this study and establish a consensus for standardizing wrist circumference measurements.

## 5. Conclusions

Wrist circumference was significantly associated with other adiposity indicators (waist circumference and BMI) and cardiometabolic risk components. The WrC cutoff points identified by the study for determining overweight were categorized by gender and age groups and can be used to identify adults with excess weight and predict cardiometabolic risk.

Wrist circumference showed a significant correlation with other adiposity indicators and can be used to identify overweight adults and predict cardiometabolic risk.

## Figures and Tables

**Figure 1 ijerph-21-00549-f001:**
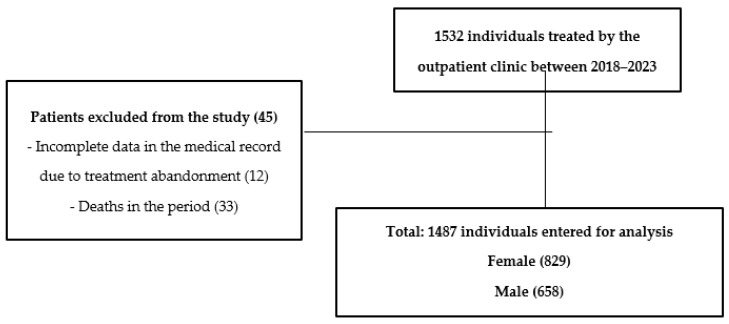
Study design.

**Figure 2 ijerph-21-00549-f002:**
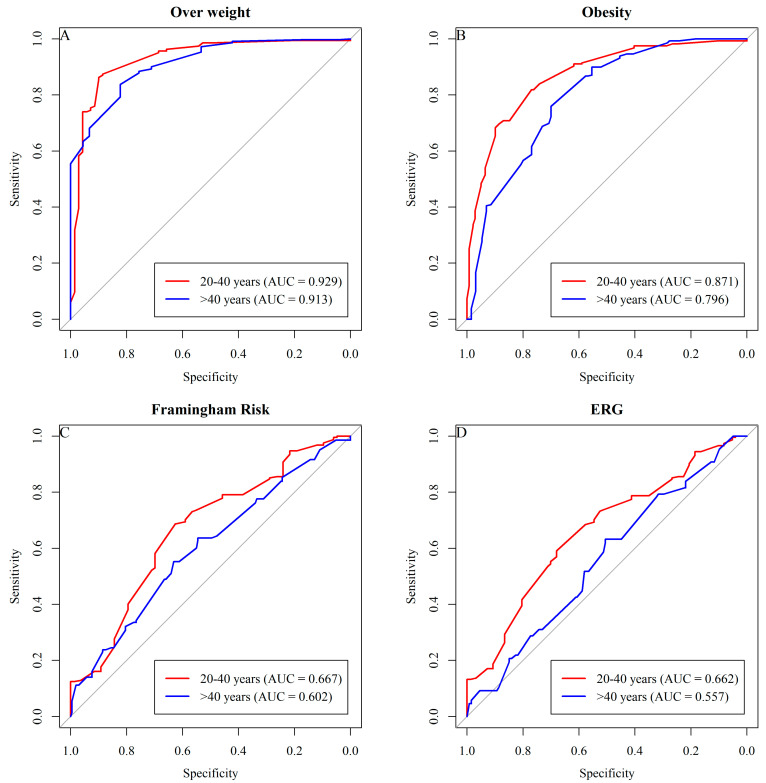
The graphical results of the ROC curve.

**Figure 3 ijerph-21-00549-f003:**
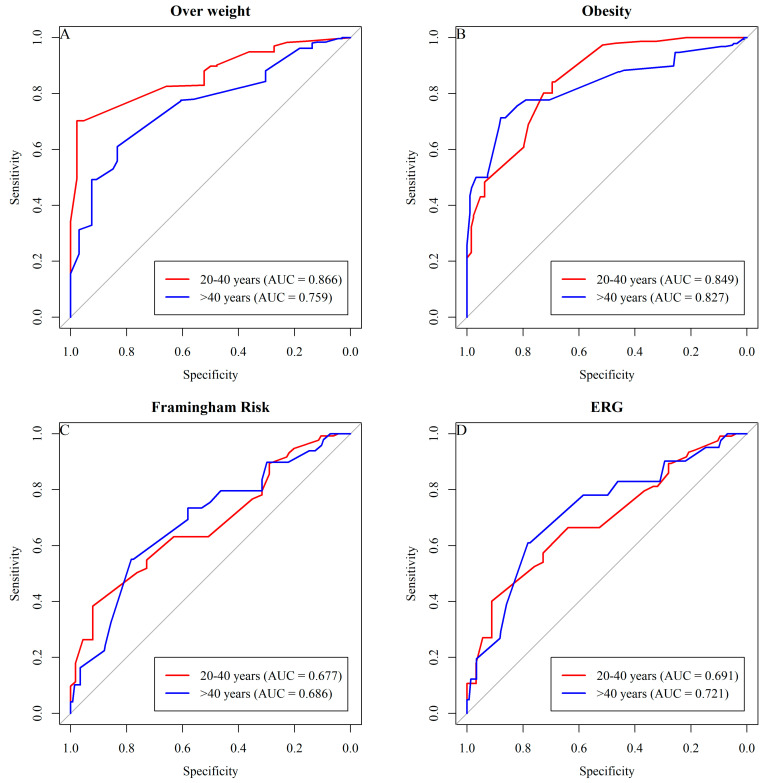
The graphical results of the ROC curve, indicate reasonable predictive quality for most cases, except for the Framingham risk in both age groups and ERG in the >40 age group.

**Figure 4 ijerph-21-00549-f004:**
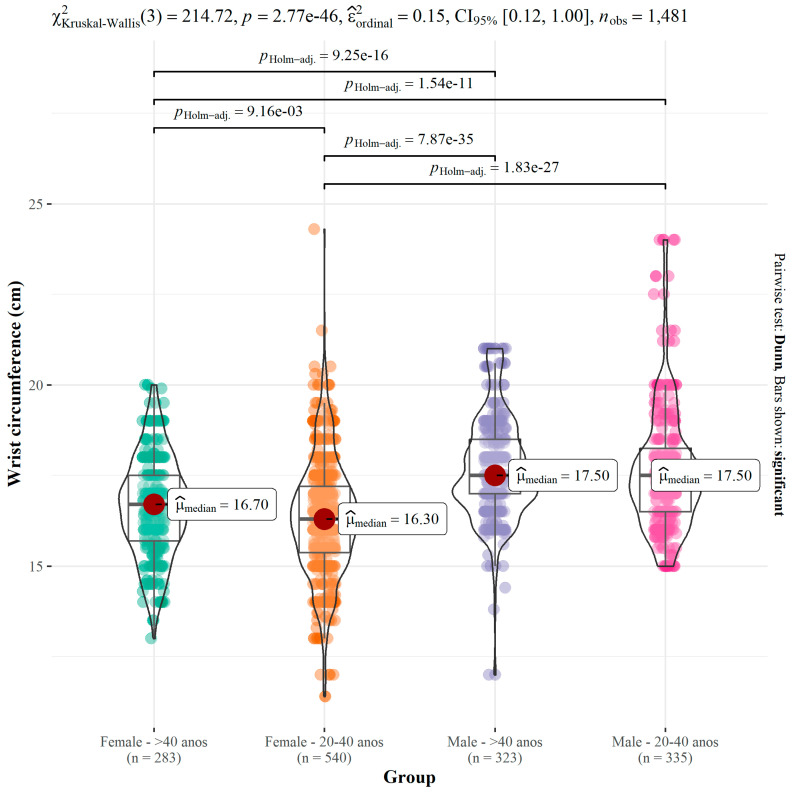
Cut-off points for Wrist Circumference (WrC) for predicting excess weight.

**Table 1 ijerph-21-00549-t001:** Demographic, socioeconomic characterization, physical activity, cardiovascular risk and nutritional diagnosis according to the age group of the sample of individuals treated at the Hospital University of Sergipe, Brazil (*n* = 1487).

Variable/Category	Age Range		*p*-Value
	20 to 40 Years *n* (%)	>40 Years Old *n* (%)	
SEX			
Feminine	420 (60.1)	409 (51.9)	0.002
Masculine	279 (39.9)	379 (48.1)	
Framingham classification			
High	185 (32)	432 (62.4)	<0.001
Intermediary	12 (2)	68 (9.9)	
Low	382 (66)	190 (27.7)	
ERG			
High	179 (30.9)	436 (63)	<0.001
Intermediary	43 (7.4)	128 (18.5)	
Low	357 (61.7)	128 (18.5)	
IPAQ			
Active	459 (65.7)	592 (75.1)	<0.001
Inactive	240 (34.3)	196 (24.9)	
BMI			
Low weight	32 (4.6)	9 (1.1)	<0.001
Eutrophy	82 (11.7)	102 (12.9)	
Overweight	153 (21.9)	210 (26.6)	
Obesity	432 (61.8)	467 (59.3)	

Legend: *n* absolute frequency; %: percentage relative frequency. ERG: Global Risk Score; IPAQ: International Physical Activity Questionnaire; BMI: Body Mass Index.

**Table 2 ijerph-21-00549-t002:** Characterization of anthropometric measurements of the study sample according to age group (*n* = 1478).

Variable	Age Range		*p*-Value
	20 to 40 Years Median (IIQ)	>40 Years Old Median (IIQ)	
Weight (kg)	87.4 (73.3–105)	84.6 (73.3–95.6)	0.002
Height (cm)	1.65 (1.59–1.72)	1.62 (1.57–1.7)	<0.001
BMI (kg/m^2^)	31.6 (27.3–39.3)	32.1 (27.7–36.8)	0.206
WrC (cm)	17 (15.8–18)	17 (16–18)	0.009
WrCE (cm)	37 (34.2–39)	37 (35–38.5)	0.855
WC (cm)	98 (86.8–110)	100 (91.9–109)	0.003
WHtR (cm)	0.59 (0.53–0.67)	0.61 (0.56–0.67)	<0.001
WHR (cm)	0.88 (0.81–0.93)	0.92 (0.86–0.97)	<0.001

Caption: IIQ—interquartile range. Caption: BMI: Body Mass Index; WrC: Wrist circumference; WrCE: Neck circumference; WC: Waist circumference; WHtR: Waist–height ratio and WHR: Waist–hip ratio; Mann–Whitney test.

**Table 3 ijerph-21-00549-t003:** Correlation between wrist circumference and anthropometric measurements of the study sample.

Variable	Wrist Circumference		
	Total R (*p*-Value)	20 to 40 Years R (*p*-Value)	>40 Years Old R (*p*-Value)
**Weight (kg)**	0.61 (<0.001)	0.64 (<0.001)	0.57 (<0.001)
**Height (cm)**	0.31 (<0.001)	0.31 (<0.001)	0.33 (<0.001)
**BMI (kg/m^2^)**	0.51 (<0.001)	0.55 (<0.001)	0.44 (<0.001)
**WrCE (cm)**	0.56 (<0.001)	0.61 (<0.001)	0.48 (<0.001)
**WC (cm)**	0.53 (<0.001)	0.57 (<0.001)	0.48 (<0.001)
**WHtR (cm)**	0.43 (<0.001)	0.46 (<0.001)	0.38 (<0.001)
**WHR (cm)**	0.29 (<0.001)	0.33 (<0.001)	0.27 (<0.001)

Spearman correlation. BMI Legend: Body Mass Index; WrC: Wrist circumference; WrCE: Neck circumference; WC: Waist circumference; WHtR: Waist to height ratio; WHR: Waist and hip ratio; IIQ—interquartile range.

**Table 4 ijerph-21-00549-t004:** Multiple linear regression between WrC, anthropometric indicators, SBP and lipid and glucose profile of the study sample.

	Wrist Circumference					
Variables	*β* (EP)	T (*p*-Value)	Bstd ^_^	VIF	R^2^	R^2^_adj_
Weight (kg)	7.12 (0.25)	28.79 (<0.001)	0.61	0.94	0.38	0.38
Height (cm)	0.72 (0.08)	8.16 (<0.001)	0.17	0.93	0.43	0.42
BMI (kg/m^2^)	2.30 (0.09)	26.10 (<0.001)	0.57	0.94	0.33	0.32
WrCE (cm)	0.84 (0.04)	23.12 (<0.001)	0.49	0.93	0.37	0.37
WC (cm)	4.86 (0.21)	23.02 (<0.001)	0.53	0.92	0.28	0.28
WHtR (cm)	0.03 (0.001)	19.90 (<0.001)	0.47	0.92	0.22	0.22
WHR (cm)	0.01 (0.001)	7.93 (<0.001)	0.19	0.92	0.24	0.24
SBP (mmHg)	1.35 (0.19)	7.26 (<0.001)	0.18	0.94	0.13	0.13
PAD (mmHg)	0.91 (0.12)	7.61 (<0.001)	0.20	0.94	0.08	0.07
Total cholesterol	0.62 (0.55)	1.15 (0.251)	0.03	0.94	0.01	0.01
LDL	0.11 (0.47)	0.24 (0.811)	0.01	0.94	0.01	0.01
HDL	0.01 (0.14)	0.07 (0.940)	0.002	0.94	0.01	0.003
Triglycerides	2.19 (1.23)	1.77 (0.076)	0.05	0.95	0.02	0.02
Fasting blood glucose	−0.66 (0.62)	−1.06 (0.290)	−0.03	0.94	0.05	0.05

Caption: Caption BMI: Body Mass Index; WrCE: Neck circumference; WC: Waist circumference; WHtR: Waist to height ratio; WHR: Waist and hip ratio; SBP: Systolic blood pressure; PAD: Peripheral arterial disease; LDL: Low Lipoprotein Density; HDL: High-Density Lipoprotein. *β*—regression coefficient. EP—Standard error. T—T Statistics. Bstd—standardized regression coefficient. VIF—Variance inflation factor. R^2^—coefficient of determination. R^2^adj—adjusted coefficient of determination. Linear regression model adjusted for age and sex.

**Table 5 ijerph-21-00549-t005:** Correlation between wrist circumference and cardiovascular risk in the study sample.

Variable	Wrist Circumference		
	Total R (*p*-Value)	20 to 40 Years R (*p*-Value)	>40 Years Old R (*p*-Value)
TOTAL CHOLESTEROL	0.07 (0.011)	0.10 (0.012)	0.04 (0.244)
LDL	0.02 (0.436)	0.03 (0.545)	0.02 (0.676)
HDL	−0.04 (0.114)	−0.02 (0.586)	−0.07 (0.061)
TRIGLYCERIDES	0.15 (<0.001)	0.25 (<0.001)	0.07 (0.078)
FASTING GLUCOSE	0.09 (0.002)	0.14 (0.002)	0.04 (0.334)

Spearman correlation.

**Table 6 ijerph-21-00549-t006:** NC cutoff points in predicting overweight and obesity in females by age group.

Variable	Circumference of Wrist (cm)	IF	ES	B.C	PPV	VPN	AUC	Youden
Overweight								
20 to 40 years	15.6	83.8	82.2	83.6	97.4	38.5	0.913	0.660
>40 years old	15.4	86.3	90	86.9	97.7	56.7	0.929	0.762
Obesity								
20 to 40 years	16.1	76.0	70	74.1	84.5	57.6	0.796	0.460
>40 years old	16	81.8	77	80.2	87.8	67.7	0.871	0.588
Framingham Risk								
20 to 40 years	16.4	64.4	52.4	64.1	55.6	61.5	0.593	0.169
>40 years old	16.6	62.6	68.7	74.0	40.0	84.6	0.667	0.313
ERG								
20 to 40 years	16.4	63.9	52.2	64.8	56.1	60.2	0.590	0.162
>40 years old	16.6	62.3	67.8	75.6	36.9	85.6	0.668	0.302

Caption: IF—inflation factor; ES—Specificity; B.C—regression coefficient. PPV—Positive Predictive Value. VPN—Negative Predictive Value. AUC—area under the curve.

## Data Availability

The datasets are available from Larissa Monteiro Costa Pereira upon reasonable request.

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
