# Peer review of "Wrist Circumference Cutoff Points for Determining Excess Weight Levels and Predicting Cardiometabolic Risk in Adults"

_ijerph, 2024, doi:10.3390/ijerph21050549_

Round 1

Reviewer 1 Report

Comments and Suggestions for Authors

The manuscript presents the results of a study aimed at determining wrist circumference cutoff points for determining overweight and predicting cardiometabolic risk in adults. Overall the manuscript is clear and well structured. However, I have the following comments and concerns:

Introduction section: please add the purpose of the study.

Materials and Methods section:

2.5 - Why did the authors choose the presented cardiometabolic riskmetrics scales?

2.8 - The statistical analysis section should be shortened by stating the methods used in analyzing the data obtained.

Results:

Table 1 - The title indicates risk factors. What did the authors mean by nutritional diagnosis? The table only presents data by group according to BMI.

Table 2 - There is no transcription of RCE in the note.

Conclusions:

Add wrist circumference cut-off point values identified in the study for men and women, age groups, as the best values to determine risk.

And also conclude that these values can be used to determine cardiometabolic risk in patients in the study region.

Author Response

Dear Reviewer
I hope everything is alright. We greatly appreciate your comments and accept all requests and remain available for any other debts.

Best Regards

Reviewer 2 Report

Comments and Suggestions for Authors

ijerph-2880060

The study aims to utilise patient record data collected between 2018 and 2023 to calculate wrist circumference cut points for identification of excess weight and cardiometabolic risk prediction. The study may provide useful data for the scientific body of knowledge, yet many areas need to be addressed prior to being considered for publication as noted below.

    Line 29: remove “based on this evidence” as the preceding sentence is not evidence
    Line 32: abbreviations must be spelt out in full at first use
    Line 34: remove sentence describing descriptive data method, not necessary for abstract
    Line 35-38: make description of data analysis methods more concise
    Line 39: define abbreviation
    Lines 44-50: amend text so that you are not repeating the same information in results and conclusion
    Line: 53: you should avoid using words that feature in your title. Use other relevant keywords

    Line 56: Disjointed sentence. break up sentence or reword
    Line 60: Poor grammar, amend wording to improve
    Lines 56-71: Overall the introduction is too brief. The section needs to be thoroughly grammar checked and more depth of critical analysis of literature regarding the utility of wrist circumference as a cardio metabolic risk factor is needed

    line 76: check grammar
    Lines 104, 108, 121: The description of methodology for data collection is too brief and non-descriptive. Please add comprehensive detail on data collection protocols
    Line 122: comprehensive description of how wrist circumference was measured must be added
    Line 135: efforts should be made in this section to improve grammar and make the overall section more concise where possible
    Line 135: rationale should be provided for why data were sub analysed using the age ranges 20-40 and >40. Why was this utilised instead of more precise groups such as 20-30, 40-50 and so on..
    Line 146: Just describe the data analysis procedures used and specifics on how you conducted them. Description of the general purpose of statistics is not necessary for this readership
    Line 190: The ethics approval information should be relocated to section 2.1 on line 78
    Line 199 and 201: in addition to p values, describe the difference between groups
    Line 210: make it more clear the meaning of p values and ensure consistency with significance reporting for each row
    Line 210: some data collected has not been reported, for example employment status
    Line 212: check punctuation
    Line 228: should CP be used to abbreviate wrist circumference here? Same for CPE and CC

Comments on the Quality of English Language

There are many errors throughout the manuscript. Please re-examine writing throughout and improve grammar

Author Response

(The authors gave the same response as above.)

Round 2

Reviewer 1 Report

Comments and Suggestions for Authors

I would like to extend my gratitude to the authors for their hard work in revising the manuscript. This has allowed them to present the results more clearly.

While I am satisfied with their answers to my questions, I feel that section 2.8, "Statistical Analysis", could be shortened further. It would be sufficient to describe the methods used for statistical analysis and the specific steps taken to perform them. In my opinion, a detailed description of the statistical analysis methods is not necessary for this audience.

Please check lines 425-428 and 429-430 to ensure this is not a repetition.

Author Response

Dear Reviewer
I hope everything is alright. Initially, I would like to thank the comments that greatly helped to improve the manuscript, and all considerations were taken into account.

Best Regards

Reviewer 2 Report

Comments and Suggestions for Authors

The author amendments have addressed my comments satisfactorily, good work

Comments on the Quality of English Language

good overall, editing services should check details finely

Author Response

(The authors gave the same response as above.)
